# Analysis of Intratumoral Heterogeneity in Myelodysplastic Syndromes with Isolated del(5q) Using a Single Cell Approach

**DOI:** 10.3390/cancers13040841

**Published:** 2021-02-17

**Authors:** Pamela Acha, Laura Palomo, Francisco Fuster-Tormo, Blanca Xicoy, Mar Mallo, Ana Manzanares, Javier Grau, Silvia Marcé, Isabel Granada, Marta Rodríguez-Luaces, María Diez-Campelo, Lurdes Zamora, Francesc Solé

**Affiliations:** 1MDS Group, Institut de Recerca Contra la Leucèmia Josep Carreras, ICO-Hospital Germans Trias i Pujol, Universitat Autònoma de Barcelona, 08916 Badalona, Spain; pacha@carrerasresearch.org (P.A.); lpalomo@carrerasresearch.org (L.P.); ffuster@carrerasresearch.org (F.F.-T.); mmallo@carrerasresearch.org (M.M.); amanzanares@carrerasresearch.org (A.M.); 2Hematology Service, ICO-Hospital Germans Trias i Pujol, Institut de Recerca Contra la Leucèmia Josep Carreras, Universitat Autònoma de Barcelona, 08916 Badalona, Spain; bxicoy@iconcologia.net (B.X.); jgrau@iconcologia.net (J.G.); smarce@iconcologia.net (S.M.); igranada@iconcologia.net (I.G.); lzamora@iconcologia.net (L.Z.); 3Microarrays Unit, Institut de Recerca Contra la Leucèmia Josep Carreras, ICO-Hospital Germans Trias i Pujol, Universitat Autònoma de Barcelona, 08916 Badalona, Spain; 4Hematology Service, ICO-Hospital Verge de la Cinta de Tortosa, 43500 Tarragona, Spain; marta.rodriguez@iconcologia.net; 5Hematology Service, Hospital Universitario de Salamanca, 37007 Salamanca, Spain; mdiezcampelo@usal.es

**Keywords:** myelodysplastic syndromes, single cell, intratumoral heterogeneity

## Abstract

**Simple Summary:**

Myelodysplastic syndromes (MDS) are a heterogeneous group of clonal hematopoietic stem cell malignancies characterized by ineffective differentiation of one or more bone marrow cell lineages. Only 50% of patients with de novo MDS will be found to have cytogenetic abnormalities, of which del(5q) is the most common. In 10% of MDS cases, del(5q) is found as a sole abnormality. In this work, a single cell approach was used to analyze intratumoral heterogeneity in four patients with MDS with isolated del(5q). We were able to observe that an ancestral event in one patient can appear as a secondary hit in another one, thus reflecting the high intratumoral heterogeneity in MDS with isolated del(5q) and the importance of patient-specific molecular characterization.

**Abstract:**

Myelodysplastic syndromes (MDS) are a heterogeneous group of hematological diseases. Among them, the most well characterized subtype is MDS with isolated chromosome 5q deletion (MDS del(5q)), which is the only one defined by a cytogenetic abnormality that makes these patients candidates to be treated with lenalidomide. During the last decade, single cell (SC) analysis has emerged as a powerful tool to decipher clonal architecture and to further understand cancer and other diseases at higher resolution level compared to bulk sequencing techniques. In this study, a SC approach was used to analyze intratumoral heterogeneity in four patients with MDS del(5q). Single CD34+CD117+CD45+CD19- bone marrow hematopoietic stem progenitor cells were isolated using the C1 system (Fluidigm) from diagnosis or before receiving any treatment and from available follow-up samples. Selected somatic alterations were further analyzed in SC by high-throughput qPCR (Biomark HD, Fluidigm) using specific TaqMan assays. A median of 175 cells per sample were analyzed. Inferred clonal architectures were relatively simple and either linear or branching. Similar to previous studies based on bulk sequencing to infer clonal architecture, we were able to observe that an ancestral event in one patient can appear as a secondary hit in another one, thus reflecting the high intratumoral heterogeneity in MDS del(5q) and the importance of patient-specific molecular characterization.

## 1. Introduction

Myelodysplastic syndromes (MDS) are a heterogeneous group of hematologic malignancies characterized by ineffective hematopoiesis which ultimately derives from peripheral blood (PB) cytopenias and dysplasia. Mainly based on those characteristics, the World Health Organization (WHO) distinguishes six MDS adult subtypes that differ on their risk of progression to acute myeloid leukemia (AML) [1]. Among them, the most well characterized subtype is MDS with isolated chromosome 5q deletion (MDS del(5q)), which is the only one defined by a cytogenetic abnormality [2]. Lenalidomide is an immunomodulatory drug that preferentially affects del(5q) cells, leading to a complete cytogenetic remission and transfusion independence in 50% and 70% of MDS del(5q) patients, respectively [3,4]. 

MDS patients are genetically heterogeneous, with approximately 90% of cases harboring recurrent somatic mutations affecting around 40 different genes [5,6,7]. Despite none of them being specific of the disease, associations between mutations and prognosis or response to treatments have been described, such as the good prognosis of *SF3B1* mutations and the association of *TP53* mutations with adverse outcomes and resistance to lenalidomide [8,9]. Moreover, several studies have documented the complexity of clonal evolution in MDS describing both linear and branched evolutionary patterns even in low risk MDS cases [10,11]. These studies also corroborate that therapy induces a selective pressure capable of reshaping mutational architecture [12,13,14].

Previous studies use conventional or bulk sequencing approaches, which average information regarding mutational pattern in a sample using DNA from admixed cell populations comprised by different tumor clones as well as normal cells. Thus, clonal architecture (CA) of the sample can be inferred using the variant allele frequency (VAF) of each detected mutation: through statistical methods and bioinformatic algorithms, mutations with similar VAF are clustered together. However, this approach provides low resolution when there is not enough difference between VAF values, as mutations cannot accurately be separated into new clones and tend to be placed together in existing ones. This can also lead to miss small and rare cell subpopulations, which are often implicated in disease progression and relapse. In the last decade, single cell (SC) analysis has emerged as a powerful tool that might overcome these difficulties, allowing a higher resolution level to study cancer and to further understand the disease [15,16]. 

Herein, we performed SC studies in four patients with MDS del(5q) at diagnosis or before receiving any treatment (DX/PRE) and, for three of them, during follow-up (FU). For this purpose, CD34+CD117+CD45+CD19- bone marrow (BM) hematopoietic stem progenitor cells (HSPC) were studied. A median of 175 cells per sample were analyzed and considered for CA inference. 

## 2. Results

### 2.1. Baseline Characteristics and Genetic Analysis of MDS Patients

Four female patients with MDS del(5q), diagnosed according to the WHO 2017 classification criteria, were included in the study [1]. Median age at DX was 78 years old (range: 69–83). Table 1 shows the main clinical and hematological characteristics of each patient at each studied time point. 

At DX/PRE, all patients presented with low hemoglobin levels, which reflects the anemia. Risk was assessed according to the Revised International Prognostic Scoring System (IPSS-R) [17], with low and very low risk for all patients. Conventional cytogenetics (CC) and fluorescence in situ hybridization (FISH) were performed by standard routine analysis. Three patients presented del(5q) as an isolated alteration while patient P3 also harbored a reciprocal translocation between both chromosomes 1.

BM and PB samples were obtained from all patients at DX/PRE and from available FU. BM samples were used to isolate bulk tumoral DNA and to sort CD34+CD117+CD45+CD19− HSPC for subsequent SC studies. Matched germline control DNA was obtained from T-CD3+ lymphocytes from DX/PRE PB samples. A general overview of the sample processing is shown in Figure 1.

Whole exome sequencing (WES) analysis and single nucleotide polymorphism arrays (SNP-A) were used to detect mutations (single nucleotide variants (SNVs) and small insertion/deletions) and copy number alterations (CNA) in DX/PRE samples, respectively. In addition, targeted deep sequencing (TDS) was performed in DNA samples from BM at DX/PRE and last available FU samples using a custom capture-based panel targeting 40 myeloid-related genes (Appendix A).

WES revealed 29 somatic mutations across the four patients at DX/PRE where non synonymous SNVs were the most common alterations (25/29 mutations) (Table 2; Appendix A). Each patient harbored a median of 7 somatic mutations and, as expected, similar VAF values were observed by WES and TDS in common analyzed genes. Analysis of FU samples by TDS revealed the same somatic mutations from the DX/PRE, except for patient P4, in which an additional *TP53* mutation was detected only in the FU sample, suggesting that it was acquired during treatment with lenalidomide (TDS was also performed at DX/PRE, confirming the absence of this mutation at this time point). Additionally, SNP-A allowed a more precise definition of the del(5q) breakpoints (Appendix A) and revealed the presence of a 1.67 Mb microdeletion in the Xp chromosome of patient P2.

Based on the results of genetic analyses (WES and SNP-A), candidate SNVs and CNAs were selected for each patient for posterior targeted qPCR analysis at SC resolution in DX/PRE and available FU samples. SNVs highlighted in bold in Table 2 and CNA detected by SNP-A were selected for subsequent SC studies.

### 2.2. Clonal Architecture in CD34+CD117+CD45+CD19- HSPC

We analyzed a median of 175 cells per sample (range: 94–201) that were considered for CA inference after analyzing sequential samples from each patient.

Proposed CAs for each patient are represented in Figure 2, Figure 3, Figure 4 and Figure 5. The number of identifiable subclones varied from one to three. Inferred CA were relatively simple and either linear or branching.

#### 2.2.1. Patient P1

A total of six alterations were included for SC studies at 3 time points: DX, FU after 14 and 33 months of DX, respectively (Table 1 and Table 2). Figure 2 shows the detected clones and proposed CA for each moment. 

At DX, a clone harboring *TP53*, *LRTOMT*, *NUP93* and del(5q) followed a linear evolutionary process and gave rise to another subclone that acquired *SETBP1* mutation. These two clones could be detected in both FU samples. Additionally, 14 months after DX, a subclone harboring *CUX1* mutation was detected. This subclone could not be detected in the sample obtained 33 months after the DX; however, another clone harboring *TP53*, *LRTOMT*, *NUP93* but without del(5q) was detected. It was inferred that this clone might precede the other ones. 

Considering that this patient was included in the placebo arm in the SINTRA-REV clinical trial, our results would be reflecting the natural course of the disease. No significant changes were observed at molecular level, nor were additional chromosomal lesions detected in the FU samples. According to the IPSS-R criteria, the patient was categorized as a very low-risk case at the moment of DX. However, considering that the hemoglobin level decreases and blast percentage increases, the IPSS-R increases in two points, recategorizing the patient in the low-risk group.

Additionally, a *TP53* mutation was present in all detected clones. There is evidence from previous studies that this mutation occurs in an early stage of the disease in at least one fifth of low-risk MDS patients with del(5q) and that it is associated with a shorter median overall survival and with an increased risk of disease evolution [18,19]. Therefore, the presence of this mutation might be contributing to clinical changes that were previously described.

#### 2.2.2. Patient P2

Five alterations were included for SC studies at 3 time points: PRE, FU after 14 and 22 months, respectively (Table 1 and Table 2). In addition to del(5q), del(Xp), detected by SNP-A, was also included in SC studies. 

Results at PRE reveal a clone harboring *SF3B1*, *YLMP1* and del(Xp) (Figure 3). Following a branched evolutionary process, this clone gave rise to two other ones: the most prevalent clone harboring del(5q) and a small subclone without del(5q) that, instead, presented with *CACHD1* mutation. In contrast to patient P1, in this case del(5q) is not present in the inferred ancestral clone at the moment of PRE. Nonetheless, the secondary clone harboring del(5q) constitutes the predominant clone in CD34+CD117+CD45+CD19- HSPC compartment at that time point. 

After 14 months of being enrolled in the SINTRA-REV clinical trial, in contrast to normal karyotype results, SC studies revealed that the clone harboring del(5q) was still present but in a significantly lower proportion (only 5% of analyzed cells). However, in the second FU sample (22 months after the initial sample), del(5q) was not detected anymore by either CC or SC analysis. This complete cytogenetic response is consistent with this patient being in lenalidomide arm in the SINTRA-REV clinical trial.

Notably, this is the only case where a homozygous mutation was detected. SNV in *YLPM1* gene was previously detected by WES at 46% VAF, but only a mutated probe signal was detected by SC analysis in tumoral cells. Considering the VAF detected by WES, the presence of a homozygous mutation was not initially expected. However, the fact that WES was performed in bulk BM DNA and SC studies were done in a HSPC subpopulation of the BM might provide an explanation for the low VAF value.

#### 2.2.3. Patient P3

A total of 7 alterations were selected for SC studies (Table 1 and Table 2). Two clones, which differed only in *TRIM24* mutation, were detected at the moment of PRE (Figure 4). Two months after this sample was obtained, patient progressed to myelofibrosis and no evaluable BM samples could be further obtained for SC studies due to BM fibrosis. Unfortunately, the patient died before another sample could be obtained. 

No complex intratumoral heterogeneity was observed in this patient. This is consistent with the fact that MDS del(5q) patients are commonly grouped as low and very low risk categories.

#### 2.2.4. Patient P4

Four alterations were selected for SC studies (Table 1 and Table 2) at 2 time points: DX and FU after 20 months.

SC analysis at DX revealed the presence of a unique clone harboring, besides del(5q), *NUP85* and *BMP7* mutations (Figure 5). Subsequently, a FU sample obtained 20 months after DX while the patient was under lenalidomide treatment, was analyzed. Results revealed that the initial clone gave rise to another one that additionally carried *TP53* mutation and that represented the dominant clone in the CD34+CD117+CD45+CD19- HSPC compartment. The presence of *TP53* mutations in MDS del(5q) is associated with lower response rates to lenalidomide, which would be consistent with the fact that the patient did not achieve cytogenetic response even when hemoglobin levels improved after treatment. 

In addition, cells with no alterations were also detected at this moment. It is highly probable that those cells correspond to a non-tumoral population (healthy HSPC). However, the existence of small CNA or other SNV cannot be fully discarded because SNP-A and WES were only performed in DX samples.

## 3. Discussion

Most of our current knowledge about intratumoral heterogeneity in MDS derives from bulk sequencing studies. However, part of this heterogeneity is the result of the coexistence of nonmalignant cells and diverse subpopulations of tumoral cells, each one harboring their own genetic characteristics, which might remain masked when bulk sequencing approaches are used. VAF values might give an idea regarding the size of a clone carrying a specific mutation and this might help to establish the order of acquisition of each mutational event when sequential samples are analyzed. Nonetheless, when there is not enough difference between VAF values, this approach could lead to mistakes when trying to resolve co-occurrence patterns of gene mutations and it is unsuccessful in resolving zygosity states. Because of this, rare cancer cells might remain masked and intratumoral complexity could be underestimated using bulk sequencing strategies [15,20]. All these difficulties might be overcome with SC strategies.

Except for two recent studies [21,22], and unlike other diseases such as AML or multiple myeloma [20,21,23,24,25,26,27], DNA SC strategies have not been applied in the MDS research field. In the present study, SC was applied to a selected group of MDS del(5q) patients.

Previous studies performed by Nilsson et al. [28,29] supported the view that del(5q) is an early event in the pathobiology of MDS del(5q), which was verified by Woll et al. in 2014 [30]. In our study, in 3 out of 4 studied patients, we could verify the presence of del(5q) in all SCs, suggesting that this alteration is present in the most ancestral clones that were detected at the moment of DX/PRE. Patient P2, in which del(5q) was detected in a secondary clone, was the exception. It is noteworthy that the clone preceding the one harboring del(5q) in patient P2 harbored a mutation in the *SF3B1* gene. According to the literature, this mutation often represents an ancestral event in MDS and it is detected in approximately 20% of MDS del(5q) [8]. This finding is in line with the results from Mossner et al., who, contrary to what was reported before, demonstrated that del(5q) constitutes a secondary event in 62% of MDS del(5q), according to their patient cohort [12]. Similarly, Woll et al. reported four cases where *SF3B1* mutation preceded del(5q), but these constituted cases of either refractory anemia with ring sideroblasts or refractory cytopenia with multilineage dysplasia. Mian et al. studied the role of *SF3B1* mutation in a cohort of patients with MDS with ring sideroblasts and concluded that such alteration might have an initial role in the pathogenesis of that MDS subtype [31]. Co-occurrence of *SF3B1* mutation and del(5q) was a matter of debate in a recent publication of the International Working Group for the Prognosis of MDS [32], in which they provided evidence supporting the recognition of *SF3B1*-mutant MDS as a distinct entity. Regarding those cases with *SF3B1* mutation and del(5q), they argued that, although genetic ontogeny of these myelodysplastic clones might inform the classification process and determine whether a case with concomitant del(5q) and *SF3B1* mutation should be more appropriately classified as MDS del(5q) or MDS with mutated *SF3B1*, in many cases, clonal hierarchy cannot be easily solved in the clinical practice. At present, these cases should be classified according to current WHO criteria within the category of MDS del(5q) [1].

To date, lenalidomide has been approved in Europe for the treatment of low-risk transfusion-dependent patients with MDS del(5q), when other therapeutic options are insufficient or inadequate [33]. Even when 56% to 67% of patients achieve transfusion independence, after 2–3 years of treatment, clinical and cytogenetic relapse might appear in 50% of cases [3,33,34,35]. In our study, patients P2, P3 and P4 received lenalidomide, while patient P1 did not receive treatment (placebo arm in the SINTRA-REV clinical trial). 

Clinical and molecular data were consistent with patient P2 being in lenalidomide arm considering that, after 14 months of being in the clinical trial, complete cytogenetic response was achieved with a slight improvement in hemoglobin levels. Surprisingly, SC studies revealed that del(5q) was still present at that FU moment in 5% of analyzed cells, despite a normal karyotype was observed by CC. However, in the sample analyzed 22 months after the diagnosis, this clone was undetectable by both CC and our SC approach. This finding highlights the elevated sensibility of SC techniques, which also depends on the number of analyzed cells. 

Regarding patient P1, del(5q) was present in all studied samples, and hemoglobin levels were gradually decreasing as the IPSS-R score increased. As previously mentioned, this is consistent with the patient being in the placebo group. Interestingly, in the last FU sample, we detected a clone without del(5q) but with *TP53* mutation suggesting that, in this patient, the mutation might precede the deletion. On the contrary, even when CC results suggest that the proportion of cells with such deletion was increasing, those results are not comparable to SC analysis because the cell population that was studied by each technique is not the same and also because real BM composition might be underrepresented by the sampling process. 

Even when it is reported that nearly 20% of patients with MDS del(5q) harbor *TP53* mutations [18], Scharenberg et al. suggested that, in some cases, this mutation is acquired during the disease evolution, especially after lenalidomide treatment [36]. This seems to correlate with patient P4’s situation, in which *TP53* mutation was only detected in the FU sample by TDS at 7% VAF, suggesting that it constituted a secondary subclonal event. This was verified by SC analysis, which revealed that the single clone that was detected at DX gave rise to another one that acquired *TP53* mutation and that, in proportion, represents the biggest clone in the FU sample. Besides, the presence of this mutation has been previously associated with lower rates of cytogenetic response [9,36], but not with lower hematologic response rates [2,12,18]. This also correlates with the clinical features of patient P4. Even when CC reveals the same number of metaphases with del(5q) in both studied samples, an improvement in hemoglobin levels was noticed (that impacted in decreasing IPSS-R score). SC analysis also revealed a pool of cells without any of the studied alterations. It is highly probable that those cells belong to a non-tumoral population, normal HSPC that might be part the pool of cells contributing to the clinical improvement mentioned before. 

Overall, few changes were observed in the CA of the studied patients. This is reasonable, considering that the studied MDS subtype is generally associated to IPSS-R low risk categories, patients often present no other cytopenia besides anemia, BM blast percentage is normally <5% and high-risk cytogenetic alterations are usually not observed [1]. In line with the aforementioned, it is common to find a low number of genetic alterations in such low risk patients in contrast to other MDS subtypes generally associated with a higher risk, such as MDS with excess of blasts [7,9]. Moreover, considering that the median overall survival of MDS del(5q) patients is around 66–145 months [1], such results could be expected, taking into account that the largest FU included sample was obtained 33 months after DX. 

During the last few years, SC techniques have rapidly evolved and part of this advance is due to its combination with sequencing techniques [37]. Most of the new high-throughput SC instruments are based on microfluidic systems, where thousands of barcoded droplets carry each SC for subsequent library generation in a single tube [20,38]. Unlike these new instruments, capture in the C1 system is limited by the number of capture sites in the chip [26,37]. As an advantage of this system, and contrary to the newest ones, with the C1 system, it is possible to verify (by microscope visualization) and select for subsequent analysis only those capture sites containing one cell. 

Our approach allowed us to study the intratumoral heterogeneity of four patients with MDS del(5q). Even when it is a limited subset of cases, compared to previous reports where larger cohorts were studied [11,12,13,30,39], clonal composition and evolution patterns were figured out in those studies based on mutation VAF values in bulk tumoral samples. Despite such differences, studies performed by Woll et al., Mossner et al. and ours were able to observe that an ancestral event in one patient can appear as a secondary hit in another one [12,30].

## 4. Materials and Methods 

### 4.1. Patients

Four patients with MDS del(5q), diagnosed according to the WHO 2017 classification criteria, were included in the study [1]. Patients were diagnosed in Hospital Institut Català d’Oncologia (ICO)–Germans Trias i Pujol (Badalona, Spain, *n* = 3) and in Hospital Verge de la Cinta (Tortosa, Spain, *n* = 1). PB and BM samples were collected after written informed consent in accordance with the Declaration of Helsinki and after the approval from Hospital Germans Trias i Pujol Ethics Committee (Reference number: PI-21-058). 

CC and FISH were performed by standard routine analysis, as previously described [40].

Patients P1 and P2 were enrolled in SINTRA-REV clinical trial (registered at clinicaltrials.gov (accessed on 21 January 2021) as NCT01243476), which is a phase III multicenter, randomized, double blind and controlled with placebo trial. SINTRA-REV was designed to assess the efficacy and toxicity of the scheme lenalidomide (5 mg qd for 28 days) versus observation in low risk MDS associated with del(5q) with anemia (Hemoglobin ≤ 12 g/dL) but without transfusion requirements. 

### 4.2. Sample Processing

BM and PB samples were obtained from all patients at DX/PRE and from available FU. BM samples obtained at DX/PRE and at available FU were used to isolate bulk tumoral DNA using the Maxwell 16 Blood DNA Purification Kit (Promega, Madison, WI, USA), and to sort CD34+CD117+CD45+CD19- HSPC with the FACSAria™ II cell sorter (BD Biosciences, San José, CA, USA) (Figure 1; Appendix A). Matched germline control DNA was obtained from T-CD3+ lymphocytes from DX/PRE PB samples. 

### 4.3. Mutation and Copy Number (CN) Analysis

WES analysis and SNP-A were used to detect mutations (SNVs and small insertion/deletions) and CNA in DX/PRE samples, respectively. WES libraries were prepared from 1.5 µg of genomic DNA using the SureSelect Human Exome Kit 51 Mb V5 (Agilent Technologies, Santa Clara, CA, USA) and sequenced on a HiSeq2500 instrument (Illumina, San Diego, CA, USA) following a standard 2 × 100 bp paired-end reads protocol at a minimum mean coverage of 140x for tumoral samples and 60× for control germline samples. WES data were analyzed using an in-house bioinformatics pipeline (Appendix A). Genomic microarrays were performed with the CytoScan® HD (Thermo Fisher Scientific, Waltham, MA, USA), following the CytoScan^TM^ User Guide (P/N 703038 Rev. 4), and analyzed with the Chromosome Analysis Suite version 3.0.0.42 (Thermo Fisher Scientific, Waltham, MA, USA) software (Appendix A). 

In addition, TDS was performed in DNA samples from BM at DX/PRE and last available FU samples using a custom capture-based panel targeting 40 myeloid-related genes (Appendix A). Libraries were performed with the SureSelectQXT Target Enrichment for Illumina Multiplexed Sequencing chemistry (Agilent Technologies, Santa Clara, CA, USA). TDS was performed on a MiSeq instrument (Illumina) following a paired-end 2 × 75 bp reads standard protocol, with a mean coverage of 1000×. Data were analyzed using a previously reported in-house bioinformatics pipeline [41]. 

Based on the results of genetic analyses, candidate mutations and CNAs were selected for each patient for posterior targeted qPCR analysis at SC resolution in DX/PRE and available FU samples. 

### 4.4. SC Isolation

SC isolation and processing were performed with the Fluidigm C1 platform (Fluidigm, San Francisco, CA, USA) using the C1 Single-Cell Open App IFC microfluidic chip for tumoral HSPC isolation (10–17 µm IFC) or control cells isolation (5–10 µm IFC; healthy donor T-CD3+ lymphocytes served as wild type and normal CN control). Cell lysis and specific targeted DNA pre-amplification are part of SC processing that take place in the C1. Designed TaqMan assays (Thermo Fisher Scientific) or custom designed LNA prime time assays (Integrated DNA Technologies, Coralville, IA, USA) were used to test selected alterations of each patient. Protocols were based on those described by the manufacturer (PN 100-6117, Fluidigm) (Appendix A).

### 4.5. SC Multiplex qPCR for Genotyping and CNA Analysis

SC analysis was performed following a previously described approach for SC multiplex qPCR analysis [23,42]. As mentioned before, individual mutation-specific genotyping assays were custom designed for mutation analysis and three different TaqMan CN assays covering each chromosomal region of interest were used for CNA analysis (Appendix A). A predesigned genotyping assay (rs346172) for loci in heterozygosis was used as a reference, while B2M locus, located in a diploid region of the genome, was used as a control for CN analysis. Additionally, BM bulk tumoral DNA from each tested patient was used as a positive control and T-CD3+ lymphocytes DNA from a healthy donor was used as a negative control. 

Genotyping assays were tested in triplicates whereas CN assays were tested in quadruplicates. The specific target amplification product (from C1 pre-amplification step) was diluted (1:5) and qPCR was performed using the 96 × 96 Gene Expression Dynamic Array and the BioMarkTM HD (Fluidigm). PCR cycling conditions are detailed in Appendix A. 

Fluidigm Real-Time PCR Analysis software v.4.3.1. was used for mutations and CN analyses and CopyCaller v2.1 (Applied Biosystems, Thermo Fisher Scientific) was used to estimate the calculated CN values. Cycle threshold (Ct) was individually established for each assay, then cells without control amplification and with Ct > 30 were discarded. A heterozygous mutation was considered to be present if the signals from the mutant and wild-type sequence probes (FAM and VIC respectively) were present in an SC. A homozygous mutation was considered to be present if wild-type sequence signal was absent. To determine the CN for selected locus, the ΔΔCt method with modifications was used to determine the relative CN for each locus, as previously described [42]. 

Results from all interrogated mutations and CNAs per cell were transformed into binary data (1, presence of mutation/CNA; 0, wild type) and combined. The total number of analyzable tumoral SCs and relative percentage of each population were calculated. The threshold to define subclonal populations was established in at least 5% of the total of analyzable tumor cells.

## 5. Conclusions

Despite the limited subset of cases, intratumoral heterogeneity of four patients with MDS del(5q) was studied using a SC approach. As far as we know, this is the first time that an SC study reflects and confirms high intratumoral heterogeneity in MDS del(5q), reinforcing the importance of patient-specific molecular characterization. Although MDS del(5q) patients do not present with a very high molecular complexity, their intratumoral heterogeneity could be more complex than conventional studies have previously shown. SC studies with larger patient cohorts and analyzing a higher number of cells are desirable in order to explore deeper the complexity of this and other hematological malignancies.

## Figures and Tables

**Figure 1 cancers-13-00841-f001:**
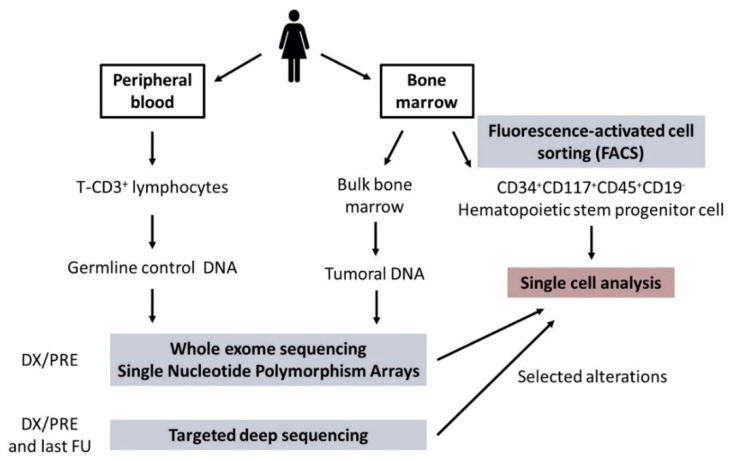
General overview of sample processing. Abbreviations: DX/PRE: diagnosis/pre-treatment; FU: follow-up.

**Figure 2 cancers-13-00841-f002:**
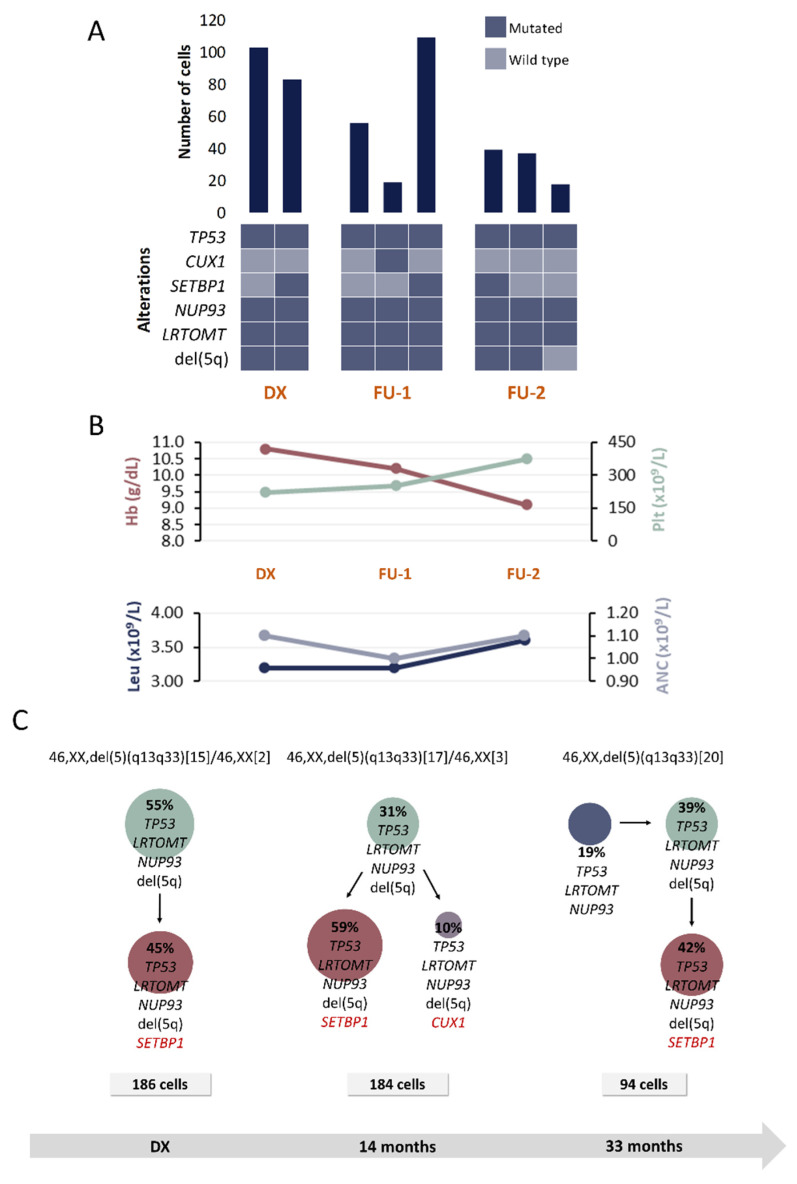
Clinical and molecular characteristics of patient P1. (**A**) Bar plot depicts the number of cells with a given genotype for each clone that was detected at the moment of diagnosis (DX), 1st follow-up (FU-1) and 2nd follow-up (FU-2). Heatmap below the bars indicates the genotype for each clone: dark blue and light blue are used to indicate the mutated or wild-type state, respectively. (**B**) Hematological characteristics such as hemoglobin (Hb), platelet count (Plt), leukocyte count (Leu) and absolute neutrophil count (ANC) are indicated for each analyzed time point. (**C**) Proposed clonal architecture, karyotype and number of analyzed cells for each time point is indicated. Color red distinguishes new acquired alterations in each clone.

**Figure 3 cancers-13-00841-f003:**
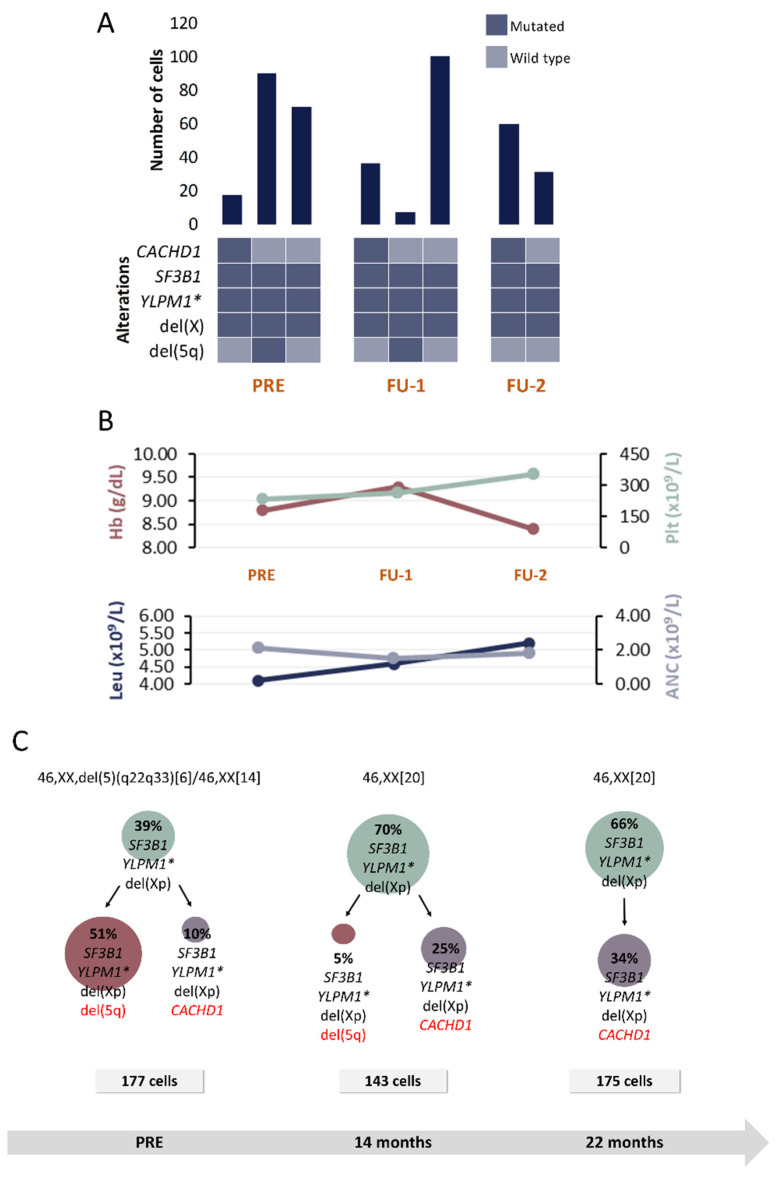
Clinical and molecular characteristics of patient P2. (**A**) Bar plot depicts the number of cells with a given genotype for each clone that was detected at the moment of pre-treatment (PRE), 1st follow-up (FU-1) and 2nd follow-up (FU-2). Heatmap below the bars indicates the genotype for each clone: dark blue and light blue are used to indicate the mutated or wild-type state, respectively. (**B**) Hematological characteristics such as hemoglobin (Hb), platelet count (Plt), leukocyte count (Leu) and absolute neutrophil count (ANC) are indicated for each analyzed time point. (**C**) Proposed clonal architecture, karyotype and number of analyzed cells for each time point is indicated. Color red distinguishes new acquired alterations in each clone. * homozygous mutation.

**Figure 4 cancers-13-00841-f004:**
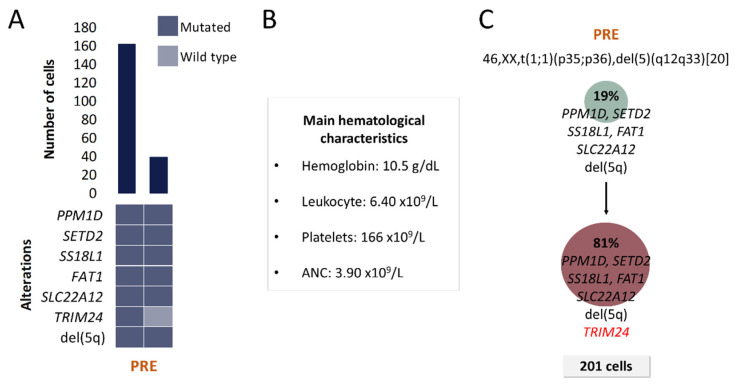
Clinical and molecular characteristics of patient P3. (**A**) Bar plot depicts the number of cells with a given genotype for each clone that was detected at the moment of pre-treatment (PRE). Heatmap below the bars indicates the genotype for each clone: dark blue and light blue are used to indicate the mutated or wild-type state, respectively. (**B**) Hematological characteristics such as hemoglobin (Hb), platelet count (Plt), leukocyte count (Leu) and absolute neutrophil count (ANC) are indicated for the analyzed time point. (**C**) Proposed clonal architecture, karyotype and number of analyzed cells for the analyzed time point are indicated. Color red distinguishes new acquired alterations in the secondary clone.

**Figure 5 cancers-13-00841-f005:**
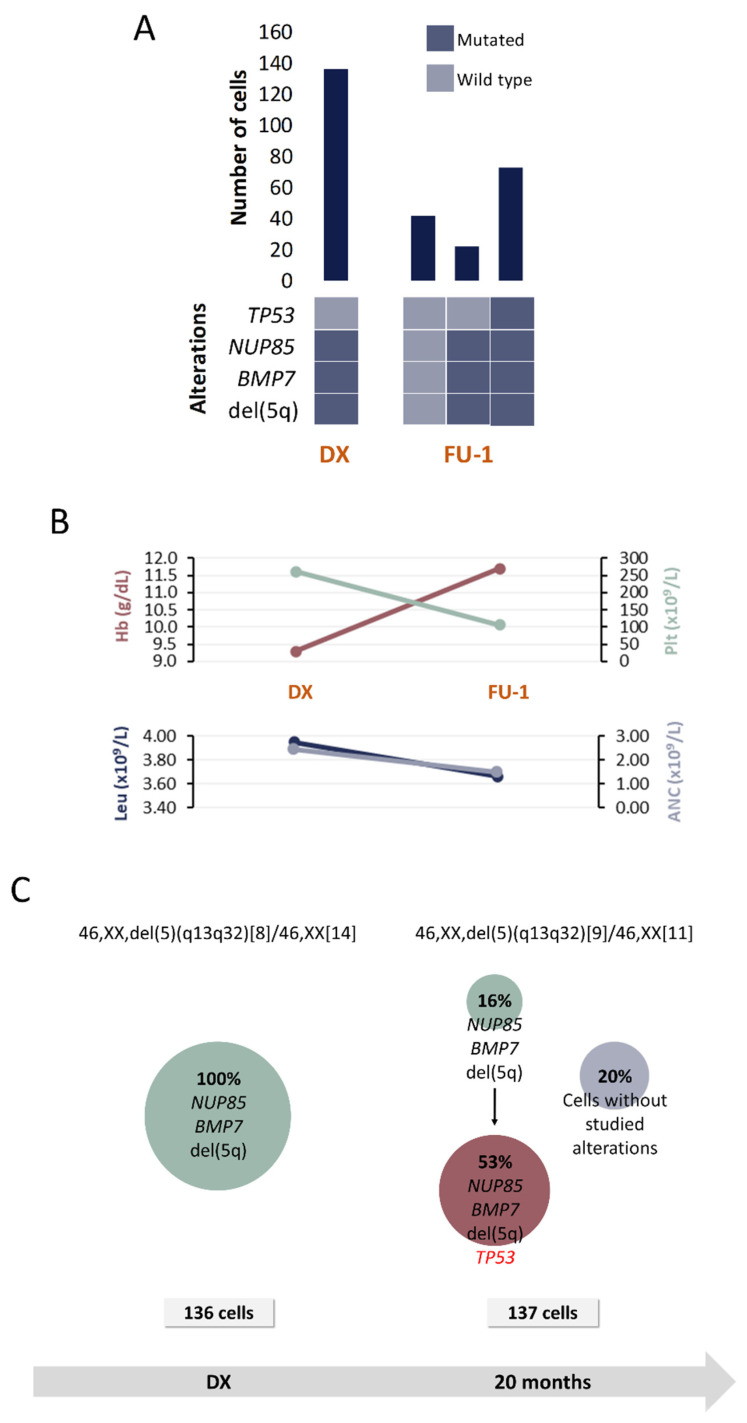
Clinical and molecular characteristics of patient P4. (**A**) Bar plot depicts the number of cells with a given genotype for each clone that was detected at the moment of diagnosis (DX), 1st follow-up (FU-1) and 2nd follow-up (FU-2). Heatmap below the bars indicates the genotype for each clone: dark blue and light blue are used to indicate the mutated or wild-type state, respectively. (**B**) Hematological characteristics such as hemoglobin (Hb), platelet count (Plt), leukocyte count (Leu) and absolute neutrophil count (ANC) are indicated for each analyzed time point. (**C**) Proposed clonal architecture, karyotype and number of analyzed cells for each time point are indicated. Color red distinguishes new acquired alterations in each clone.

**Table 1 cancers-13-00841-t001:** Main clinical and hematological data of the four patients included in the study.

UPN	Age at DX and Sex	Sampling Time Points	Hb (g/dL)	WBC (×10^9^/L)	Platelets (×10^9^/L)	ANC (×10^9^/L)	BM Blasts (%)	Karyotype, SNP-A and FISH (ISCN)	IPSS-R	Treatment
P1	79 Female	DX	10.8	3.20	221	1.10	1.0	46,XX,del(5)(q13q33)[15]/46,XX[2] arr[hg19] 5q14.3q34(89575437-163450743)x1 nuc ish(D5S1518Ex2,D5S1976x2,EGR1x1,RPS14x1)[63/100]	1 Very low	SINTRA-REV (Placebo)
		1st FU (14 months after diagnosis)	10.2	3.20	252	1.00	0.0	46,XX,del(5)(q13q33)[17]/46,XX[3] nuc ish(D5S1518Ex2,D5S1976x2,EGR1x1,RPS14x1)[80/100]	1 Very low	
		2nd FU (33 months after diagnosis)	9.1	3.60	374	1.10	3.0	46,XX,del(5)(q13q33)[20] nuc ish(D5S1518Ex2,D5S1976x2,EGR1x1,RPS14x1)[88/100]	3 Low	
P2	69 Female	PRE: 4 years after diagnosis (never received treatment)	8.8	4.10	233	2.10	0.5	46,XX,del(5)(q22q33)[6]/46,XX[14] arr[hg19] 5q21.2q34(102986652-162755919)x1, Xp22.31(6449753-8135644)x1 FISH: NA	2 Low	SINTRA-REV (Lenalidomide)
		1st FU (14 months after diagnosis)	9.3	4.60	262	1.52	0.5	46,XX[20] nuc ish(D5S1518E,D5S1976,EGR1,RPS14)x2[100]	2 Low	
		2nd FU (22 months after diagnosis)	8.4	5.20	353	1.80	1.0	46,XX[20] FISH: NA	2 Low	
P3	77 Female	PRE: 2 years after diagnosis (previously: support treatment)	10.5	6.40	166	3.90	1.5	46,XX,t(1;1)(p35;p36),del(5)(q12q33)[20] arr[hg19] 5q14.3q34(86255729-166126310)x1 FISH: NA	2 Low	Lenalidomide
P4	83 Female	DX	9.3	3.95	262	2.44	3.0	46,XX,del(5)(q13q32)[9]/46,XX[11] arr[hg19] 5q21.3q34(107937392-165840296)x1 nuc ish(D5S1518Ex2,D5S1976x2,EGR1x1,RPS14x1)[32/100]	3 Low	Lenalidomide
		1st FU (20 months after diagnosis)	11.7	3.66	106	1.47	4.0	46,XX,del(5)(q13q32)[9]/46,XX[11] nuc ish(D5S1518Ex2,D5S1976x2,EGR1x1,RPS14x1)[29/100]	2 Low	

FISH: XL 5q31/5q33/5p15 locus-specific deletion probe was used to detect 5q deletions (D-5081-100-TC, MetaSystems Probes). SINTRA-REV: phase III multicenter, randomized, double blind and controlled with placebo clinical trial and with two arms designed to assess the efficiency and toxicity of the scheme Lenalidomide versus observation in a series of 60 patients with low risk myelodysplastic syndrome associated to 5q deletion with anemia (Hb ≤ 12 g/dL) but without the need of transfusion. Abbreviations: ANC: absolute neutrophil count; BM: bone marrow; DX: diagnosis; FISH: fluorescence in situ hybridization; FU: follow-up; Hb: hemoglobin; IPSS-R: Revised International Prognostic Scoring System; NA: not available; PRE: pre-treatment; SNP-A: single nucleotide polymorphism arrays; UPN: unique patient number; WBC: white blood cells.

**Table 2 cancers-13-00841-t002:** Single nucleotide variants (SNVs) detected by whole exome sequencing (WES) and targeted deep sequencing (TDS) in diagnosis/pre-treatment samples and in the last available FU sample.

UPN	Gene	Chr	Transcript	Type of Alteration	Variant	Aminoacid Change	VAF (%)		
							WES DX/PRE	TDS DX/PRE	TDS FU
**P1**	***CUX1***	**7**	**NM_001202543**	**Stopgain SNV**	**c.3019C>T**	**p.Arg1007 ***	**3**		
	***SETBP1***	**18**	**NM_015559**	**Non-synonymous SNV**	**c.2612T>C**	**p.Ile871Thr**	**15**	**14**	**17**
	*MAP7D2*	X	NM_001168465	Non-synonymous SNV	c.116A>G	p.Asn39Ser	24		
	*TENM1*	X	NM_014253	Non-synonymous SNV	c.3857G>T	p.Cys1286Phe	27		
	***LRTOMT***	**11**	**NM_001145310**	**Non-synonymous SNV**	**c.671G>A**	**p.Arg224His**	**34**		
	*CCDC168*	13	NM_001146197	Non-synonymous SNV	c.122A>C	p.Gln41Pro	34		
	***TP53***	**17**	**NM_001126115**	**Non-synonymous SNV**	**c.448C>T**	**p.Arg150Trp**	**39**	**38**	**42**
	***NUP93***	**16**	**NM_001242795**	**Non-synonymous SNV**	**c.473C>A**	**p.Ala158Asp**	**40**		
	*UNC79*	14	NM_020818	Non-synonymous SNV	c.7055T>A	p.Val2352Glu	44		
**P2**	*LRRC45*	17	NM_144999	Non-synonymous SNV	c.1670A>G	p.Glu557Gly	7		
	*CRIPAK*	4	NM_175918	Non-synonymous SNV	c.25A>C	p.Asn9His	7		
	***CACHD1***	**1**	**NM_020925**	**Non-synonymous SNV**	**c.1161G>T**	**p.Arg387Ser**	**9**		
	*IL21R*	16	NM_181078	Non-synonymous SNV	c.179A>G	p.Asp60Gly	27		
	***SF3B1***	**2**	**NM_012433**	**Non-synonymous SNV**	**c.2098A>G**	**p.Lys700Glu**	**40**	**39**	**43**
	***YLPM1***	**14**	**NM_019589**	**Stopgain SNV**	**c.4087C>T**	**p.Arg1363 ***	**46**		
**P3**	*CGNL1*	15	NM_001252335	Non-synonymous SNV	c.3481C>A	p.Arg1161Ser	7		
	*IBSP*	4	NM_004967	Non-synonymous SNV	c.231G>T	p.Glu77Asp	8		
	***SLC22A12***	**11**	**NM_144585**	**Non-synonymous SNV**	**c.232C>T**	**p.Pro78Ser**	**14**		
	***TRIM24***	**7**	**NM_015905**	**Non-synonymous SNV**	**c.263A>G**	**p.Tyr88Cys**	**14**		
	***SETD2***	**3**	**NM_014159**	**Non-synonymous SNV**	**c.6197A>G**	**p.Asp2066Gly**	**21**		
	***FAT1***	**4**	**NM_005245**	**Non-synonymous SNV**	**c.1507G>A**	**p.Ala503Thr**	**23**		
	*TCHH*	1	NM_007113	Non-synonymous SNV	c.3770A>G	p.Gln1257Arg	28		
	***PPM1D***	**17**	**NM_003620**	**Stopgain SNV**	**c.1434C>A**	**p.Cys478 ***	**29**		
	***SS18L1***	**20**	**XM_005260390**	**Non-synonymous SNV**	**c.604A>G**	**p.Ser202Gly**	**29**		
**P4**	***TP53***	**17**	**NM_001276761**	**Non-synonymous SNV**	**c.283T>C**	**p.Phe95Leu**	**ND**	**ND**	**7**
	*SCUBE1*	22	NM_173050	Non-synonymous SNV	c.1700C>T	p.Ala567Val	8		
	***BMP7***	**20**	**NM_001719**	**Non-synonymous SNV**	**c.908G>A**	**p.Arg303His**	**14**		
	***NUP85***	**17**	**NM_024844**	**Non-synonymous SNV**	**c.877G>C**	**p.Ala293Pro**	**17**		
	*DNMT3B*	20	NM_006892	Splicing variant	c.1906-1G>T	NA	21		

SNVs in bold were selected for single cell analysis. Abbreviations: Chr: chromosome; DX: diagnosis; FU: follow-up; ND: not detected; PRE: pre-treatment; SNV: single nucleotide variant; TDS: targeted deep sequencing; UPN: unique patient number; VAF: variant allele frequency; WES: whole exome sequencing. * is to indicate that a stop codon is introduced.

## Data Availability

The data presented in this study are available on request from the corresponding author.

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
