# Peer review of "Analysis of Intratumoral Heterogeneity in Myelodysplastic Syndromes with Isolated del(5q) Using a Single Cell Approach"

_cancers, 2021, doi:10.3390/cancers13040841_

Round 1

Reviewer 1 Report

In this manuscript, the authors address the high intratumoral heterogeneity in four patients with MDS with isolated del(5q) by a single cell analysis approach.It is a very interesting, and the experimental results are presented in an easy-to-understand manner. The system of single cell analysis is capable of observing clonal changes over time. However, I have several concerns that need to be addressed before considering publication.

First of all, the number of cases is so small that it is difficult to obtain universal conclusions.

Second, it is not clear what clinical utility this method of genetic analysis has for a patient with MDS del(5q).

Eventually, it is not clear what the advantage and clinical usefulness of this single cell analysis compared to the conventional methods of genetic analysis.

So, this paper provides very interesting data but it still needs a considerable revision to be acceptable for the Cancers Journal.

Reviewer 2 Report

The authors present an interesting study that uses a single cell approach to assess intratumoral heterogeneity in 4 patients with myelodysplastic syndromes with a del(5q). The manuscript is well written, and the figures are informative. I have the following concerns.

  1. Given the relatively low cell number (94-204), how do the authors make sure that the studied cells are representative for the clonal distribution?
  2. Related to above, the summary states: “We were able to demonstrate that an ancestral event in one patient can appear as a secondary hit in another one”. This is a strong statement and additional evidence is needed to support it. Is the secondary hit independent of the ancestry event in the sample?
  3. Related to 2) What is the relation of the observed mutations to known hot spots?
  4. Related to 3) Annotation is not provided for the 29 mutations – do they have a COSMIC ID or rsID, and what is (if any) their population frequency?

Technical:

  1. The number and the type of analyzed cells should be stated in the abstract/intro.
  2. Line 113 – “in all four patients” does not sound correct – do the authors mean “across the 4 patients”?

Reviewer 3 Report

In this work  Dr. Acha and coworkers employed  a single-cell targeted approach  to study  intratumoral heterogeneity in four patients with MDS del(5q). More in detail, they firstly employed whole exome sequencing (WES) analysis and single nucleotide polymorphism  arrrays (SNP-A) to detect mutations (SNVs and small insertion/deletions) and copy number alterations in samples at diagnosis. Subsequently, candidate SNVs and CNAs selected for each patient for  analysis on single  CD34+CD117+CD45+CD19- bone marrow cells at diagnosis and from available follow-up samples were analyzed by by high-throughput qPCR and clonal architecture was inferred.

This study confirms at leats in part what currently accepted about the early origin of the del(5q) lesion in MDS and provides interesting observations on the timing of TP53 mutation. On a general note, this experimental work (on a limited subset of samples) illustrates the advantages of a single cell approach applied to "longitudinal" sampling as compared to bulk sequencing in determining the ITH. This study confirms that, within the del(5q) group, a significant ITH does exist and can be traced by SC analysis

This work rely on a limited subset of samples: however, it is well organized and follows an appropriate logical workflow. The text is clear and the methods appropriately described. The figures and the legends may need some improvement.

In general the  figures are a bit crowded and may be a bit confusing. The text is not easy to read (especially referred to the percentage of mutations). The legend should be made  more explanatory and should follow more closely  the logical flow of the panels. Pheraps the use of lettters to enumerate the panels in figs 2-5 will help.

Round 2

Reviewer 1 Report

This paper discusses the importance of single cell techniques in the assessment of minimal residual disease in myelodysplastic syndrome (MDS) with isolated del(5q). Although the number of cases is so small, the techniques of single cell analysis might be contributing to a better understanding of the MDS with del(5q). Further studies are needed to verify the clinical usefulness of single cell analysis. However, I recommend that this paper is worth publishing.

Reviewer 2 Report

The authors addressed most of my concerns.